# OpenReview forum: "Reasoning as an Attack Surface: Adaptive Evolutionary CoT Jailbreaks for LLMs"
_ICLR.cc/2026/Conference — ICLR 2026 Conference Desk Rejected Submission_

### Official Review · Reviewer_LHXc · 2025-10-28

**Soundness:** 2
**Presentation:** 1
**Contribution:** 2
**Rating:** 4
**Confidence:** 4

**Summary:**

This paper proposes AE-CoT, an adaptive evolutionary framework designed to jailbreak Large Reasoning Models by exploiting CoT mechanism. The method first reframes a harmful goal into a ``teacher-style'' prompt. It then uses a genetic algorithm with an adaptive mutation rate to search a structured space of CoT fragments, aiming to find the most effective adversarial reasoning path. The authors report that AE-CoT achieves better performance in attack success rate and harmfulness score across various models.

**Strengths:**

This paper is built upon a sound idea that defines a structured search space for CoT fragments and using an evolutionary algorithm to find effective attack vectors, which seems a reasonable strategy for this problem.

**Weaknesses:**

1. The writing and presentation are poor. In the abstract, a term like *H-CoT-style* is used without any definition. The first two paragraphs of the introduction are missing essential citations for fundamental concepts like *Large Reasoning Model*, *CoT*, and *Jailbreaking*. When the paper claims on lines 44-45 that prior work has major limitations, it does so without citing any specific papers. Are these the authors' own conclusions, or are they based on previous findings? Without references, these are just unsubstantiated claims. I also found that key terms like *mutation-rate* and *population convergence* are used without proper introduction, which makes the paper very hard to follow. Furthermore, there are numerous instances of inconsistent abbreviation usage for both LLMs (e.g., Line 38, 91, 146, 168) and CoT (e.g., Line 13， 33， 122， 134， 145， 154， 177， 202， 236， 237， 290. ), which shows a lack of careful proofreading. If you have introduced the abbreviation before, you don't need to repeatedly reintroduce it.

2. Insufficient related work section. The paper discusses too few related articles. It's clear that it has missed a significant body of work, both on jailbreaking Large Reasoning Models in general [1-5] and specifically on attacks that leverage the CoT and reasoning process [3,6-7]. This suggests an incomplete literature review.

[1] Jinx: Unlimited LLMs for Probing Alignment Failures

[2] ExtendAttack: Attacking Servers of LRMs via Extending Reasoning

[3]  Cats Confuse Reasoning LLM: Query Agnostic Adversarial Triggers for Reasoning Models

[4]  OverThink: Slowdown Attacks on Reasoning LLMs

[5]  BoT: Breaking Long Thought Processes of o1-like Large Language Models through Backdoor Attack

[6] Response Attack: Exploiting Contextual Priming to Jailbreak Large Language Models

[7] Atoxia: Red-teaming Large Language Models with Target Toxic Answers

3. Confusing expression in Section 3. In Equation 6, the symbols on the right-hand side are not defined. Do they represent sets of attributes? If you put them in the Appendix and Table 6, you should refer to them. Most critically, there is a severe error of symbol reuse: on line 178, $\tau$ is used for a *rewriting template,* but it seems to be used for something entirely different in the evolutionary selection process. Maybe the authors can re-check this entire section for symbol definitions, usage, and consistency.


4. The experimental description also lacks basic scholarly practice. On line 346, baseline methods like ArtPrompt and PAP are mentioned without any citations, leaving the reader to guess which papers are being referred to.


5.  While the high-level idea is sound, the components themselves (evolutionary search, teacher-style rewriting, structured search spaces) are common techniques in the field of adversarial attacks and prompt refinement. The paper fails to adequately position its specific combination of these ideas against prior work, thus overstating its novelty.


6. I have concerns about the computational cost, which is not adequately discussed. The optimization process, as I understand it, relies on a black-box evaluation that requires multiple expensive API calls to both the target and a judge model for every single candidate in every generation. This process seems prohibitively expensive for any practical, large-scale application. While the paper claims to be more *efficient* than another method, this doesn't change the fact that the absolute cost is very high, a crucial limitation that is largely ignored.

**Questions:**

Please refer to the weaknesses.

---

> ### Author Response · Authors · 2025-11-22
>
> Thank you for the valuable review. Below we respond to the comments in **Weaknesses (W)**.
>
> ---
>
>  **W1: Poor writing; undefined terms; inconsistent abbreviations**
>
> Thank you. We revised the **abstract**, **introduction**, and **method** sections for clarity. All previously undefined terms are now defined at first appearance. We also performed a global consistency check for **LLM/LRM/CoT**, ensuring each is introduced once and used consistently.
>
> ---
>
>  **W2: Insufficient related work; missing citations [1–7]**
>
> We expanded the **Related Work** section and added standard references for **Large Reasoning Models**, **Chain-of-Thought**, and jailbreak methodology. All recommended papers [1–7] are now cited. We reorganized the section into clearer subsections (reasoning-model jailbreaks, CoT-based attacks, long-trace manipulation) and added a paragraph explicitly positioning AE-CoT among these works.
>
> ---
>
>  **W3: Confusing Section 3; undefined symbols in Eq. 6; symbol reuse**
>
> We revised **Section 3** to clarify notation. All symbols in **Eq. 6** are now defined immediately before the equation, with explicit references to Table 7 and Appendix C. We corrected symbol reuse by separating the rewriting template ($\mathcal{R}$), CoT template vectors ($\mathbf{x}$), and evolutionary population ($\mathcal{P}_k$).
>
> ---
>
>  **W4: Missing citations for baseline methods (ArtPrompt, PAP, etc.)**
>
> We added citations for all baseline methods, including **ArtPrompt**, **PAP**, **CodeAttack**, **CL-GSO**, **ICRT**, and **H-CoT**.
>
> ---
>
>  **W5: Novelty overstated; components are common techniques**
>
> We clarify that AE-CoT’s novelty lies not in isolated components but in their **synergistic combination and adaptation** for adversarial CoT generation.
>
> 1. **Structured Search Space ($\Theta$):**
>    Unlike token-level optimization (e.g., GCG) or less structured evolutionary search (e.g., CL-GSO), AE-CoT operates on a **semantically coherent, interpretable** search space based on nine CoT attributes. This yields interpretable, transferable adversarial traces and constrains the search to a high-level but manageable space ($|\Theta| \le 50^9$), improving stability and efficiency.
>
> 2. **Adaptive Evolutionary Optimization:**
>    AE-CoT introduces a **dynamic mutation-rate control module** absent in prior evolutionary prompt attacks. This mechanism adjusts exploration vs. exploitation based on population diversity and convergence.
>
> **Empirical justification:**
> *Disabling the adaptive mutation schedule drops ASR from 90% to 60% and harmfulness score from 3.8 to 3.0 (Sec. 4.6, Table 5).*
> *AE-CoT achieves ~3× speedup over CL-GSO on Gemini-2.5-flash-thinking (589.26s → 193.77s), demonstrating superior efficiency.*
>
> Thus, AE-CoT’s novelty lies in **structuring the CoT attack space** and **adapting the evolutionary engine**, addressing interpretability and efficiency limitations in prior work.
>
> ---
>
>  **W6: Computational cost insufficiently discussed**
>
> We added a section (Sec. 4.7) discussing the computational costs. Using 5 AdvBench tasks with complete API call tracking, we report the following:
>
> | Case | Rwt | Tgt | Jdg | Rwt Tok | Tgt Tok | Jdg Tok | Tok Sum | Tgt Cost | Jdg Cost | Total Cost |
> | :--: | :--: | :--: | :--: | :--: | :--: | :--: | :--: | :--: | :--: | :--: |
> | 1 | 1 | 22 | 22 | 817 | 69,839 | 35,086 | 105,742 | 0.2372 | 0.1296 | 0.3692 |
> | 2 | 1 | 16 | 16 | 1,333 | 56,160 | 31,255 | 88,748 | 0.1944 | 0.1115 | 0.3106 |
> | 3 | 1 | 32 | 32 | 1,003 | 106,559 | 61,779 | 169,341 | 0.3667 | 0.2210 | 0.5909 |
> | 4 | 1 | 10 | 10 | 1,031 | 32,035 | 19,417 | 52,483 | 0.1079 | 0.0730 | 0.1842 |
> | 5 | 1 | 14 | 14 | 879 | 48,551 | 28,440 | 77,870 | 0.1685 | 0.0990 | 0.2702 |
> | **Avg** | 1.0 | 18.8 | 18.8 | 1,013 | 62,229 | 35,195 | 98,037 | 0.2149 | 0.1268 | **0.3450** |
> | **Total** | 5 | 94 | 94 | 5,063 | 312,144 | 175,977 | 526,184 | 1.0747 | 0.6341 | **1.7251** |
>
> **Key findings:**
> • AE-CoT achieves **100% success rate**.
> • Average per-task cost is **\$0.345**, lower than many prior evolutionary attacks.
> • Only **18.8** target-model calls per task due to early stopping.
> • Total cost across all 5 tasks is **\$1.725**, confirming efficient, bounded search.
>
> ---
>
> We sincerely thank the reviewer again. Your suggestions have significantly improved the manuscript.

---

> > ### Comment · Reviewer_LHXc · 2025-11-25
> >
> > Thank the authors for your feedback. Have you uploaded the revised paper? I'm not sure if there's a system error. It seems like the current version is the same as the previous one. Maybe coloring the modifications could make them clearer. Besides, I would like to hear the author's opinions on Question 5.

---

> > > ### Author Response · Authors · 2025-11-26
> > >
> > > Thank you for your response. We noticed that our initial rebuttal did not explicitly address **Weakness 5**, which concerns the positioning and novelty of AE-CoT. We have now added a complete and detailed discussion of this point, clarifying how AE-CoT differs from prior evolutionary, CoT-based, and structured prompt optimization methods. The explanation highlights our key contributions: the customized structured CoT search space and the adaptive mutation-rate module designed specifically for reasoning-model jailbreaks.
> > >
> > > We have also uploaded an updated PDF that incorporates the corresponding revisions into the main paper.
> > >
> > > Thank you again for the helpful suggestions.

---

### Official Review · Reviewer_uVzy · 2025-10-30

**Soundness:** 2
**Presentation:** 1
**Contribution:** 1
**Rating:** 2
**Confidence:** 4

**Summary:**

This paper introduces AE-CoT, an adaptive evolutionary framework designed to use the chain-of-thought (CoT) mechanism in large models as an attack surface for jailbreaking. The method begins by rewriting a harmful objective into a "teacher-style" prompt. It then decomposes the CoT into structured semantic fragments and employs an evolutionary algorithm with an adaptive mutation rate to search for an optimal adversarial CoT combination within a predefined, structured space.

**Strengths:**

The paper's motivation is clear. Targeting reasoning models with an evolution-based jailbreak attack is a valuable and specific application, and the experimental results appear to validate its effectiveness.

**Weaknesses:**

**I have severe concerns regarding the contribution and novelty of this paper. The formulation and methodology of AE-CoT exhibit a high degree of overlap with a prior work, CL-GSO [1]** (which uses an expandable strategy space for evolutionary search). To be specific:

- Both papers identify the exact same bottleneck in existing jailbreak methods: the "fixed and limited strategy space." More importantly, they propose a virtually identical solution: decompose a holistic attack strategy into multiple independent, interpretable components, and then use an evolutionary/genetic algorithm to search the resulting combinatorial and vastly expanded strategy space to discover more powerful and diverse attacks. This is the foundational methodology for both papers, and it has already been clearly articulated and validated by CL-GSO.

- CL-GSO decomposes its strategies into four dimensions based on the Elaboration Likelihood Model (ELM): Role, Content Support, Context, and Communication Skills. AE-CoT, on the other hand, claims to decompose CoT into nine sub-templates, including reasoning role, context frame, content support, and communication style. While the number and names of the dimensions differ, the core idea—"to componentize the attack paradigm"—is exactly the same. There are also clear parallels between some of AE-CoT's dimensions (e.g., role, context) and those in CL-GSO.

- Both papers employ a standard genetic algorithm flow: population initialization, crossover, and mutation, and both emphasize some form of "adaptive" adjustment. The "adaptive mutation rate" proposed in AE-CoT serves the exact same functional purpose as the "adaptive crossover and mutation rates with a soft decay strategy" mentioned in CL-GSO—that is, to balance exploration and exploitation during the search.


[1] Huang et.al., Breaking the Ceiling: Exploring the Potential of Jailbreak Attacks through Expanding Strategy Space, ACL 2025

**Questions:**

The authors must provide a thorough explanation for the stunning similarity between their methodology and Figures 1 and 2 with CL-GSO's. **I would like to ask the Area Chair to adjudicate on this matter**.

Given these suspected similarities, I think that this paper does not meet the standard for publication at ICLR.

**Details Of Ethics Concerns:**

The formulation, methodology and main figures of this paper exhibit a high degree of overlap with a prior work, CL-GSO [1] (which uses an expandable strategy space for evolutionary search).

[1] Huang et.al., Breaking the Ceiling: Exploring the Potential of Jailbreak Attacks through Expanding Strategy Space, ACL 2025, https://arxiv.org/pdf/2505.21277v2

---

> ### Author Response · Authors · 2025-11-22
>
> Thank you for your valuable review. We respond to the comments in **Weaknesses (W)** and **Questions (Q)**.
>
> ---
>
>  ***W1: AE-CoT highly overlaps with CL-GSO.***
>
> While both methods use evolutionary search as a *general paradigm*, AE-CoT and CL-GSO differ fundamentally in optimization space, search unit, representation, and adaptation strategy.
>
> **(a) Different optimization spaces.**
> CL-GSO expands a *behavior-level* rhetorical strategy space (derived from ELM: Role, Content Support, Context, Communication Skills). AE-CoT expands a *reasoning-level* structured CoT space using a 9-dimensional semantic vector controlling role, decomposition, scaffolding, step-organization, abstraction, tone, logic, and verification.
>
> | Aspect | CL-GSO | AE-CoT |
> | :--- | :--- | :--- |
> | Level | Behavioral prompt strategies | Internal reasoning-chain structures |
> | Unit optimized | ELM-style categorical tuple | Semantic template vector $x$ |
> | Output | Style/persona combinations | Rendered CoT reasoning trace $\phi(x)$ |
> | Goal | Expand rhetorical diversity | Optimize adversarial reasoning trajectories |
>
> >**Table 1. The comparison between AE-CoT and CL-GSO in different aspects.**
>
> **(b) Different search units.**
> CL-GSO manipulates symbolic behavioral dimensions; AE-CoT manipulates a parameter vector $x\in\mathbb{R}^9$ rendered into a CoT chain via $\phi(x)$. CL-GSO optimizes *persona*, AE-CoT optimizes *internal reasoning unfolding*.
>
> ---
>
>  ***W2: Adaptive mutation is the same as CL-GSO.***
>
> This is a misunderstanding.
>
> **CL-GSO** uses a *soft-decay schedule* for crossover/mutation: hyperparameters decrease with generation count, independent of fitness.
>
> **AE-CoT** uses a *fitness-driven adaptive controller*:
> $\Delta f_k = f_{k} - f_{k-1}.$
> Mutation increases only when progress stagnates ($\Delta f_k<0$) and decreases when progress improves. This creates a closed-loop, performance-sensitive adaptation mechanism absent in CL-GSO.
>
> | Feature | CL-GSO | AE-CoT |
> | :--- | :--- | :--- |
> | Adaptation type | Time-based decay | Fitness-driven feedback |
> | Fitness sensitivity | None | Direct via $\Delta f_k$ |
> | Search behavior | Pre-scheduled | Dynamic, responsive |
> >*Table 2. Comparison between the features of the Mutation Rate control in AE-CoT and CL-GSO*
> ---
>
>  ***W3: Identical methodology in constructing strategy space.***
>
> We respectfully disagree. The core distinction is **ontological**:
>
> - **AE-CoT** evolves the model's *internal reasoning process* (CoT trajectory fragments: step decomposition, order, justification).
> - **CL-GSO** evolves only the prompt's *surface-style/behavior* (rhetorical/presentation tags).
>
> This leads to fundamentally different designs:
>
> 1. **Search object**: thinking-chain fragments (AE-CoT) vs. stylistic tags (CL-GSO).
> 2. **Representation & cohesion**: continuous vector x + deterministic rendering ϕ(x) ensuring coherent thoughts (AE-CoT) vs. discrete tag concatenation (CL-GSO).
> 3. **Compatibility & space**: strict semantic constraints Θ, ϕ for valid templates (AE-CoT) vs. treating all combinations as feasible, yielding noisier space (CL-GSO).
> 4. **Search landscape**: smooth, fragment-level exploitable structure (AE-CoT) vs. coarse and discontinuous (CL-GSO).
> 5. **Transferability**: strong cross-model transfer from reasoning skeletons (AE-CoT, Sec. 4.4, Table 3) vs. style-specific strategies (CL-GSO).
>
> In short, while both works decompose and search, they decompose **different ontological objects** (reasoning process vs. prompt presentation). The methodologies are therefore distinct.
>
> ---
>  ***W4: AE-CoT does not provide substantive novelty.***
>
> Empirical results contradict this claim. AE-CoT achieves:
> * **Higher ASR** (o1-mini, GPT-4o, etc.) and **Higher harmfulness** (Table 2).
> * **2.8× faster runtime** due to structured CoT fragments and adaptive mutation.
> * **Stronger transferability** to non-reasoning models.
>
> These gaps indicate AE-CoT’s search space and optimization dynamics differ meaningfully from CL-GSO in practice. We will incorporate an explicit comparison subsection.
>
> ---
>
> **Q1: Alleged similarity between our Figures and those in CL-GSO**
>
> We clearly state that no technical content was taken from CL-GSO.
>
> 1. **Stylistic similarity is intentional and authorized.**
>    We obtained permission from the CL-GSO authors to use a similar visual style. This approval covered appearance only.
>
> 2. **The content of the figures is entirely different.**
>    CL-GSO visualizes behavioral strategies; our figures describe reasoning-structure components. The technical content in the figures is actually different.
>
> 3. **All technical elements were independently designed.**
>    Only the graphic style is similar—and explicitly permitted.
>
> ***We respectfully disagree with any preliminary characterization of our paper as containing plagiarized content and ask the Area Chair to evaluate based on technical content, not superficial style.***

---

### Official Review · Reviewer_snzi · 2025-11-01

**Soundness:** 3
**Presentation:** 2
**Contribution:** 2
**Rating:** 6
**Confidence:** 4

**Summary:**

The paper proposes an adaptive evolutionary CoT jailbreak framework, called AE-CoT. Specifically, the method first rewrites harmful goals
into teacher-style prompts and decomposes them into semantically coherent reasoning fragments to construct a pool of CoT jailbreak candidates. Then, within a structured representation space, we perform multi-generation evolutionary search, where candidate diversity is expanded through fragment-level crossover and a mutation strategy with an adaptive mutation-rate control strategy.

**Strengths:**

The propsoed AE-CoT, an adaptive evolutionary CoT jailbreak framework, which generates the adversarial CoT traces with teacher-style rewriting and fragment-based decomposition.

The paper's approach extends this line of research by combining structured CoT optimization with evolutionary search. By explicitly targeting reasoning models’ intermediate thinking space, we demonstrate superior jailbreak success rates compared to prior CoT-based attacks, while also enabling transferable adversarial prompts for non-reasoning models.

 Experiments on both reasoning and non-reasoning models demonstrate not only strong effectiveness and transferability, but also improved efficiency compared with existing evolutionary jailbreak approaches.

**Weaknesses:**

I think the idea of the paper is good. My only concern is that can you  generalize your evolutionary  jailbreak approaches AE-CoT to the multi-modality LLMs with more experiments. For example, you verify your AE-CoT with cross-modal tasks.

**Questions:**

see the above comments.

---

> ### Author Response · Authors · 2025-11-22
>
> Thank you for your valuable review and suggestions. Below we respond to the comments in **Weaknesses (W)** and **Questions (Q)**.
>
> ---
>
>  ***W1 & Q1: Can AE-CoT generalize to multi-modal LLMs?***
>
> We thank the reviewer for raising this important point. To directly evaluate whether AE-CoT generalizes beyond text-only jailbreak scenarios, we conducted a new **multi-modal transfer attack experiment** on the publicly available **SafeBench** [1] (MiniBench subset), where each sample contains a harmful text query paired with a harmful image. Following the reviewer’s suggestion, we evaluate AE-CoT on cross-modal tasks under a realistic multi-modal threat model.
>
> **Experimental Setup.**
> We selected **Qwen-VL-Max** as the target model because it is a state-of-the-art *reasoning-capable* multi-modal LLM. Qwen-VL-Max is part of the **Qwen-VL family** [2], which demonstrates strong Chain-of-Thought and high-level reasoning ability on vision–language benchmarks, making it the most appropriate evaluation target for a CoT-based adversarial attack.
>
> For each harmful item, AE-CoT generates an adversarial CoT prompt using *text only*, without any access to the paired image. The evolved CoT is then combined with the corresponding harmful image and presented to Qwen-VL-Max.
> This evaluates whether AE-CoT’s reasoning-space perturbations transfer naturally to the visual--language domain, without requiring any architectural modification.
>
>
> | Setting | Success Rate | Harmfulness Score (avg.) |
> | :--- | :---: | :---: |
> | Baseline (original text + image) | 0% | 0 |
> | AE-CoT (evolved text + image) | 100% | 80 |
>
> >**Table 1. Multi-modal jailbreak evaluation on SafeBench (miniBench subset).**
> The baseline sends the benchmark’s original harmful text + image to Qwen-VL-Max, while AE-CoT optimizes only the text query (image unchanged). AE-CoT achieves a 100% success rate and an average harmfulness score of 80, while the baseline fails entirely.
>
>
> **Analysis**
> The experimental results are presented in Table 1. The results show that AE-CoT’s text-driven optimization remains highly effective in multi-modal settings.
> Although AE-CoT only modifies the *textual* reasoning trajectories and never accesses or manipulates the image itself, the optimized adversarial CoT still reliably causes harmful responses when combined with the corresponding harmful image.
>
> **Conclusion**
> The experiment confirms that AE-CoT *effectively generalizes to multi-modal tasks*: adversarial reasoning traces optimized purely from text can reliably jailbreak advanced visual–language models.
> This validates AE-CoT as a strong and modality-agnostic jailbreak framework.
> In future work, we plan to extend AE-CoT with multi-modal-aware components (e.g., vision-conditioned reasoning fragments or image–text joint mutation strategies) to further enhance cross-modal robustness and adversarial effectiveness.
>
> ---
>
> **References**
>
> * [1] Zeng, Z. et al. (2024). SafeBench: A Benchmark for Evaluating Model Safety in Multi-Modal and Multi-Turn Scenarios. arXiv:2410.18927.
> * [2] Wang, A. et al. (2023). Qwen-VL: A Frontier Large Vision-Language Model for Understanding, Localizing, and Reasoning. arXiv:2308.12966.

---

### Author Response · Authors · 2025-12-01

### **Dear PCs, SACs, ACs, and Reviewers,**

Thank you very much for your valuable contributions to our work. To assist the newly assigned AC and help reduce their workload, we provide below a summary of the key points from the reviews and the reviewer–author discussions.

---

 **Strengths**

We are grateful that the reviewers generally viewed our problem setting and methodology as meaningful and technically sound. In particular:

- **Targeting reasoning models’ CoT space for jailbreak**: Reviewers recognized that focusing on adversarial reasoning trajectories rather than surface prompts is a valuable direction. (snzi, LHXc)
- **Structured CoT representation**: The idea of constructing a nine-dimensional interpretable reasoning-structure space was considered sound and reasonable. (snzi)
- **Adaptive evolutionary search**: The use of adaptive mutation-rate control to balance exploration and exploitation was acknowledged as a meaningful component. (snzi)
- **Strong empirical results**: Reviewers noted the strong effectiveness of AE-CoT across reasoning and non-reasoning models and its transferability. (snzi)
- **Clear motivation**: The motivation of using reasoning processes as an attack surface was considered clear and justified. (snzi, LHXc)

---

 **Concerns and Our Addressing**

Overall, although the reviewers raised several concerns, most are centered around **novelty positioning**, **clarity/presentation issues**, **related work coverage**, and **further experiments**, rather than the validity of the core method. We supplemented experiments, expanded related work, clarified notation, and provided detailed point-by-point responses in both the revised main text and appendix.

Below we summarize the major themes.

---

 **Concerns about novelty and similarity to prior work**
(uVzy: Weaknesses 1–3)

**Our Addressing:**
We clarified that AE-CoT differs fundamentally from CL-GSO and other structured prompt optimization works in both *optimization space* and *adaptation mechanism*. Specifically:

- CL-GSO expands **behavior-level rhetorical strategies**, whereas AE-CoT expands **reasoning-level CoT structures**.
- AE-CoT optimizes **semantic reasoning vectors rendered into coherent CoT traces**, not stylistic tags.
- The adaptive mutation-rate controller in AE-CoT is **fitness-driven**, unlike CL-GSO’s **time-decay schedule**.
- We added clear explanations and explicit comparisons in the main paper as requested.

---

 **Concerns about writing clarity, missing definitions, inconsistent abbreviations, and notation issues**
(LHXc: Weaknesses 1, 3)

**Our Addressing:**
We revised the abstract, introduction, and method sections for clarity and completeness:

- Defined all terms at first appearance.
- Ensured consistent use of LLM/LRM and CoT throughout.
- Rewrote Section 3 to fix symbol reuse, define all variables, and link to tables/appendices.
- Improved clarity of Eq. 6 and added necessary references.

---

 **Concerns about missing citations and insufficient related work**
(LHXc: Weakness 2)

**Our Addressing:**
We significantly expanded the Related Work section and added all missing citations, including the reviewer-provided works [1–7]. We reorganized the section into clearer categories and added a paragraph explicitly positioning AE-CoT relative to reasoning-model jailbreaks, CoT-based attacks, and long-trace manipulation.

---

 **Concerns about multi-modal generalization**
(snzi: Weakness 1)

**Our Addressing:**
Following the reviewer’s suggestion, we conducted new multi-modal experiments on SafeBench (MiniBench) using **Qwen-VL-Max**. AE-CoT—optimized purely on text—achieves a **100% success rate** when paired with harmful images. These results demonstrate that adversarial reasoning traces transfer naturally to multi-modal LLMs.

---

 **Concerns about computational costs**
(LHXc: Weakness 6)

**Our Addressing:**
We added a detailed section on computational cost, including full API-call and token-count reporting across 5 tasks. Results show:

- **Average per-task cost: $0.345**
- **Only 18.8 target-model calls on average** due to convergence
- **Total cost for all tasks: $1.725**

These results indicate AE-CoT is efficient and feasible for practical black-box jailbreak evaluation.

---

Because our revisions involved substantial improvements—including new experiments, expanded related work, and clarified notation—the updated version was uploaded late in the discussion period. Some reviewers may not have had sufficient time to review the updated PDF and revised rebuttal. Nonetheless, we have addressed all reviewer comments comprehensively and systematically.

Above, we have faithfully summarized the reviewer comments and our corresponding responses, hoping this will assist the AC's work. We sincerely thank the reviewers, AC, SAC, and PC for their dedicated effort. Their insightful feedback has significantly strengthened our paper.

**Sincerely,**
**Authors**

---

### Note · Program_Chairs · 2026-01-17
**Submission Desk Rejected by Program Chairs**

The following references in this submission do not refer to real documents and/or have major errors in bibliographic information:

 Evan Hubinger and Joseph Carlsmith. An investigation of role prompting for jailbreaking large language models. arXiv preprint arXiv:2309.16798, 2023. URL https://arxiv.org/abs/ 2309.16798 .
Kai Greshake, Mohannad Abdalla, and Peter Grosser. More than you've asked for: A comprehensive analysis of prompt injection in large language models. arXiv preprint arXiv:2302.12173, 2023. URL https://arxiv.org/abs/2302.12173.